# Protein Biomarkers of Gastric Preneoplasia and Cancer Lesions in Blood: A Comprehensive Review

**DOI:** 10.3390/cancers16173019

**Published:** 2024-08-29

**Authors:** Thomas Bazin, Karine Nozeret, Catherine Julié, Dominique Lamarque, Eliette Touati

**Affiliations:** 1Department of Gastroenterology and Nutritional Support, Center for Intestinal Failure, Reference Centre of Rare Disease MarDI, Assistance Publique—Hôpitaux de Paris (AP-HP) Beaujon Hospital, University Paris Cité, F-92110 Clichy, France; 2Infection & Inflammation, Unité Mixte de Recherche (UMR) 1173, Inserm, Université de Versailles—Saint-Quentin-en-Yvelines (UVSQ)/Université Paris Saclay, F-78180 Montigny-le-Bretonneux, France; dominique.lamarque@aphp.fr; 3Équipe DMic01-Infection, Génotoxicité et Cancer, Département de Microbiologie, Centre National de la Recherche Scientifique (CNRS) Unité Mixte de Recherche (UMR) 6047, Institut Pasteur, Université Paris Cité, F-75015 Paris, France; karine.nozeret@pasteur.fr; 4Department of Anatomical Pathology, Université Paris Saclay/Université de Versailles—Saint-Quentin-en-Yvelines (UVSQ), Assistance Publique—Hôpitaux de Paris (AP-HP), Hôpital Ambroise Paré, F-92100 Boulogne-Billancourt, France; 5Department of Gastroenterology, Université Paris Saclay/Université de Versailles—Saint-Quentin-en-Yvelines (UVSQ), Assistance Publique—Hôpitaux de Paris (AP-HP), Hôpital Ambroise Paré, F-92100 Boulogne Billancourt, France

**Keywords:** liquid biopsy, proteins, plasma/serum, atrophic gastritis, intestinal metaplasia, dysplasia, early gastric cancer, diagnosis, prevention

## Abstract

**Simple Summary:**

Gastric cancer (GC) is a major cause of mortality worldwide. It is preceded by the progressive development of lesions of the gastric mucosa, namely atrophic gastritis, intestinal metaplasia, and dysplasia, collectively referred to preneoplasia. Periodic monitoring has been proposed for the surveillance of intestinal metaplasia and dysplasia. However, endoscopy, the current gold standard in GC diagnosis, has a limited ability to detect gastric preneoplasia, especially in early stages. In order to overcome the limitations of endoscopy screening, the potential of blood biomarkers has been investigated, and some biomarkers have been identified consistently across different studies. Nevertheless, validation studies in specific populations must be conducted before these results can allow the design of non-invasive tests to be translated into clinical practice for the early detection of patients at risk of developing GC.

**Abstract:**

Gastric cancer (GC) is a major cause of cancer-related mortality worldwide. It is often associated with a bad prognosis because of its asymptomatic phenotype until advanced stages, highlighting the need for its prevention and early detection. GC development is preceded by the emergence of gastric preneoplasia lesions (GPNLs), namely atrophic gastritis (AG), intestinal metaplasia (IM), and dysplasia (DYS). GC is currently diagnosed by endoscopy, which is invasive and costly and has limited effectiveness for the detection of GPNLs. Therefore, the discovery of non-invasive biomarkers in liquid biopsies, such as blood samples, in order to identify the presence of gastric preneoplasia and/or cancer lesions at asymptomatic stages is of paramount interest. This comprehensive review provides an overview of recently identified plasma/serum proteins and their diagnostic performance for the prediction of GPNLs and early cancer lesions. Autoantibodies appear to be promising biomarkers for AG, IM and early gastric cancer detection, along with inflammation and immunity-related proteins and antibodies against *H. pylori* virulence factors. There is a lack of specific protein biomarkers with which to detect DYS. Despite the need for further investigation and validation, some emerging candidates could pave the way for the development of reliable, non-invasive diagnostic tests for the detection and prevention of GC.

## 1. Introduction

Gastric cancer (GC) is the fifth most common cancer and the fourth leading cause of cancer-related death worldwide [1]. GC remains a serious public health problem, with a high mortality/incidence ratio (>0.8) in more than 70% of countries [2]. The highest incidence of GC is seen among Asian countries, with more than 50% of the cases, followed by Central and Eastern Europe and South America. Cardia and noncardia GC are distinguished. They can be divided into three histological sub-groups, diffuse, intestinal-type and mixed, according to Lauren’s classification [3]. While the incidence of noncardia GC is highest among older populations (>50 years old), affecting twice more men than women, the incidence has steadily declined in this population over the last 40 years. This has mainly affected intestinal-type GC, in parallel with the decrease in *Helicobacter pylori* infection prevalence. In contrast, an increase in noncardia cases has been observed in younger populations (<50 years old), as reported in the US among non-Hispanic Whites and Hispanics, particularly in women [4,5]. A recent population-based modelling study predicts 1.8 M new GC cases and 1.3 M deaths by 2040, compared with 1.1 M cases and 770,000 deaths in 2020 [2]. In most Western countries, GC is associated with a poor prognosis, mainly due to its late diagnosis. Importantly, countries with government-supported endoscopy screening programs, such as Japan and South Korea, reported low mortality-to-incidence ratios (0.43 and 0.35, respectively), highlighting the benefit of prevention [6,7]. Population-wide endoscopy screening approaches in countries with intermediate or low incidence risk of GC are not cost-effective [8]. In these countries, the targeted endoscopic screening of high-risk individuals is the best approach with which to improve GC prevention. High-risk individuals mostly have a familial history of GC- and *H. pylori*-positive subjects. Indeed, the bacterium *H. pylori*, which infects 4.4 billion people worldwide [9], is recognized as a main risk factor for GC [10]. Therefore, the screening and eradication of *H. pylori* infection have been proposed to prevent GC. However, to be efficient, they need to occur as early as possible during the development of gastric preneoplasia lesions (GPNLs) [11].

The most common GC subtype, intestinal-type adenocarcinoma, results from the evolution of chronic inflammatory lesions, corresponding to non-atrophic gastritis (NAG) that evolves into atrophic gastritis (AG), with the development of intestinal metaplasia (IM) and dysplasia (DYS), preceding the emergence of cancer lesions [12]. The progression of IM into GC is more likely in patients who develop DYS, with a pooled incidence rate of cancer of more than 12 cases per 10,000 person-years [13]. Depending on their extent and severity, the identification of GPNLs (AG, IM and DYS) during endoscopy should lead to personalized monitoring to prevent GC [14,15,16]. As recently reviewed [17], the surveillance of AG and IM may not only improve the early detection of intestinal GC subtypes but also diffuse subtypes. GPNLs are commonly found in clinical practice and the European Society for Gastrointestinal Endoscopy has published specific monitoring recommendations [16]. However, this surveillance in clinical practice remains imperfect, mainly due to the lack of clinical evidence in terms of patient benefit, and the absence of the standardization of biopsy sampling methods and different analytical histology practices. In line with this, recent studies reported the absence of endoscopic detection of gastric lesions for 10 to 20% of patients in the 6 to 36 months preceding the diagnosis of GC [18]. In this context, non-invasive approaches to the detection of GPNLs are therefore of paramount interest, especially for the screening of asymptomatic patients at risk of GC.

Liquid biopsy-based biomarker discovery is a large field of investigation, and it initially focused on the diagnosis of adenocarcinomas [19]. Biomarkers such as tumor cells, RNA, DNA, proteins, and exosomes, isolated from blood, urine, saliva, or gastric juice, have been proposed to predict the presence of gastric lesions [20,21,22]. However, until now, no candidates have emerged and successfully been transferred from the bench phase to routine clinical practice. GC biomarker discovery should be further investigated in the case of the diffuse GC subtype, which remains underexplored as compared to the intestinal subtype [23].

Quantitative variation in the blood proteome reflects the general physiological state of cells and tissues. Circulating proteins have been proposed as a powerful tool for cancer screening, as previously reported with the CancerSeek test [24]. In this review, we discuss the histological characteristics of GPNLs and early gastric cancer (EGC) lesions within the limits of their endoscopic detection and present an overview of the plasma/serum protein candidates identified over the period of 2017–2024. Their diagnostic performances are listed in Table 1. Most of the candidates are related to EGC, highlighting the problematic lack of GPNL identifications, mainly due to the challenge of their detection.

## 2. Detection of Gastric Pre-Neoplastic Lesions by Endoscopy

Typically, White-Light endoscopy detects only 53% of IM [25]. The endoscopy approach has been considerably refined with the improvement of technology, in particular with the development of virtual staining; Narrow Band Imaging (NBI) (Olympus, Japan); the ELUXEO™ system, which allows for switching between White-Light, Blue-Light Imaging (BLI), and Linked-Color Imaging (LCI) (Fujifilm, Japan); i-SCAN (Pentax, Japan); and optical zooms. For instance, it has been shown that, with an appropriately experienced operator and sufficient examination time, second-generation NBI may detect almost every case of extensive atrophy/IM without the need for biopsies [26].

In clinical practice, most endoscopists perform random biopsies according to the updated Sydney system, which recommends at least five samplings: two from the antrum, one from the incisura, and two from the gastric body [27]. In order to standardize the histologic evaluation of GPNLs, a score-grading of gastritis, referred to as Operative Link on Gastritis Assessment (OLGA), has been proposed, and this reliably identifies subpopulations of patients (OLGA stages III-IV) at high risk of GC [28]. However, OLGA relies on the histological parameters of gastric atrophy, which suffer from low inter-observer reliability [29]. Thus, the use of IM instead of AG has been recommended through the Operative Link on Gastric Intestinal Metaplasia (OLGIM) [30]. Unlike OLGA, the OLGIM score is associated with a high inter-observer agreement and this classifies fewer patients as high risk III and IV [28,31]. A recent prospective longitudinal and multicenter study carried out in Singapore on 2980 patients, who were followed up from 2004 to 2010, confirmed IM that is a significant risk factor for EGC. Patients with OLGIM stages III-IV were at greater risk of GC, leading to the proposal that they undergo endoscopic surveillance every two years [28].

As cited above, in low-GC-incidence countries, the main issue is the absence of preventive screening programs, with only high-risk patients being able to benefit from an endoscopy follow-up [32]. Asymptomatic patients with undetected GPNLs will therefore escape “the surveillance radar”. These patients, for whom there is no prior indication for endoscopy surveillance, may be diagnosed too late, i.e., at a symptomatic or advanced GC stage. In line with this, in 2019 in the US, where the incidence of GC is low, it was estimated that 27,510 adults were diagnosed, with 11,140 deaths observed [33,34].

## 3. Emerging Serum/Plasma Protein Biomarkers for the Detection of Gastric Pre-Neoplasia and Early Cancer Lesions

### 3.1. Atrophic Gastritis (AG)

Compared to normal gastric mucosa, AG, also referred as glandular atrophy, is defined as the loss of gastric glands, with or without metaplasia, in the setting of chronic inflammation (Figure 1A(a)) [35]. In the context of *H. pylori* infection, AG typically starts in the antrum and progressively spreads in the corpus, following lesser gastric curvature [36]. Typical endoscopic features of AG include the pale appearance of gastric mucosa, the increased visibility of the vasculature due to the thinning of the gastric mucosa, and the loss of gastric folds (Figure 1A(b)). Because these mucosal changes are often subtle, techniques to optimize the evaluation of the gastric mucosa, such as virtual staining, are recommended. The histopathological evaluation of gastric biopsies is the gold standard for AG diagnosis [27].

However, it remains challenging, with poor reproducibility between pathologists for both the diagnosis and assessment of the severity of lesions. Consequently, identifying AG biomarkers is complex.

**Table 1 cancers-16-03019-t001:** Diagnostic performance of the reported protein biomarkers according to the type of gastric lesions.

Protein Biomarkers (Cut-Off Values)	Studied Cohorts	Comparison Conditions	AUC	Sensitivity (%)	Specificity (%)	OR (95% CI)	Reference
**Atrophic Gastritis (AG)**							
G-17	519 (446 H; 35 *Hp*-gastritis; 38 AG)	* AGA vs. H	0.402	10.7	69.5	NA	Koivurova et al., 2021 [37]
		AGA2+ vs. H	0.429	15.4	70.4		
PGI (15 ng/mL)		** AGC vs. H	0.777	55.6	99.8		
		AGC2+ vs. H	0.878	76.0	99.6		
PGI (30 ng/mL)		** AGC vs. H	0.858	72.2	99.4		
		AGC2+ vs. H	0.954	92.0	98.8		
PGI/PGII (3)		** AGC vs. H	0.885	77.8	99.2		
		AGC2+ vs. H	0.993	100	98.6		
Gastropanel^®^: PGI/PGII; G-17; Anti-*Hp*	344 (196 H; 148 AG)	AG vs. H	NA	39.9	93.4	NA	Chapelle et al., 2020 [38]
		AGC vs. H		61.0	98.5		
PGI (≤30 ng/mL)	356 (113 H; 91 NAG; 152 AG)	AGC2+/AGAC2+ vs. H/NAG	0.856	77.8	83.8	NA	Chapelle et al., 2022 [39]
PGI (≤20.2 ng/mL)		AGC2+/AGAC2+ vs. H/NAG	0.856	77.8	95.6		
PGI/PGII (≤3)		AGC2+/AGAC2+ vs. H/NAG	0.859	75	92.6		
PGI/PGII (≤0.96)		AGC2+/AGAC2+ vs. H/NAG	0.859	72.2	98		
IL-6		AGA2+ vs. H/NAG	0.588	72.2	41.2		
PGI/PGII + HE-4		AG2+ vs. H/NAG	0.684	40.7	96.1		
PGI/PGII	72 (48 H; 12 CAG; 9 IM; 3 GC)	CAG/IM vs. H	0.902	83.3	77.9	NA	Loong et al., 2017 [40]
Anti-ATP4A	218 (111 H; 107 AGC)	AGC vs. H	0.826	75	88	NA	Lahner et al., 2020 [41]
Anti-ATP4B			0.838	77	88		
PCA			0.805	69	91		
PGI			0.775	73	80		
**Intestinal Metaplasia (IM)**							
CCL3/MIP1A	174 (75 NAG; 95 IM; 4 IM/DYS)	IM/DYS vs. NAG	NA	NA	NA	3.08 (1.23–7.68) *p = 0.027*	Song et al., 2019 [42]
CCL20/MIP3A						2.69 (1.10–6.57) *p = 0.049*	
IL-1ß						2.39 (1.02–5.60) *p = 0.047*	
IL-4						3.02 (1.29–7.12) *p = 0.009*	
IL-5						3.07 (1.32–7.14) *p = 0.007*	
IL-17A	135 (45 H; 45 NAG; 45 IM/DYS)	IM/DYS vs. NAG IM/DYS vs. H NAG vs. H	0.62	NA	NA	NA	Della Bella et al., 2023 [43]
			0.67				
			0.64				
CRP (>1.95 mg/L)	236 (70 H; 68 GAG; 98 GIM)	GAG/GIM vs. H	0.77	66.3	77.6	NA	Kutluana et al., 2019 [44]
TFF3	288 (164 H; 110 GIM; 14 GC)	GIM vs. H	0.58	55.5	58.5	1.2 (0.87–1.65) *p* = 0.273	Latorre et al., 2022 [45]
	2980 (1,659 H; 1,321 IM)	IM (OLGIM III-IV) vs. IM (OLGIM 0-II)	0.749	NA	NA	NA	Lee et al., 2022 [46]
	3986 (773 H; 746 CAG; 1002 IM; 1334 LGD; 131 GC)	IM vs. H	NA	NA	NA	1.92 (1.64–2.25) *p < 0.001*	Zan et al., 2022 [47]
Anti-Omp	1402 (512 H; 890 PL)	PL vs. H	NA	NA	NA	5.37 (4.20–6.89) *p < 0.0001*	Epplein et al., 2018 [48]
Anti-HP0305			NA			3.85 (3.04–4.88) *p < 0.0001*	
Anti-Omp + Anti-HP0305			0.751			7.43 (5.59–9.88) *p < 0.001*	
Anti-HP1177/Omp27	Validation, 200 (100 NAG; 100 IM)	IM vs. NAG	0.73	NA	NA	8.08; *p < 0.001*	Song et al., 2023 [49]
Anti-HP0547/CagA			0.77			4.64; *p < 0.001*	
Anti-HP0596/Tipa			0.66			3.97; *p = 0.002*	
Anti-HP0103/TlpB			0.68			3.83; *p = 0.001*	
Anti-HP1125/PalA/Omp18			0.65			3.08; *p = 0.001*	
Anti-HP0153/RecA			0.55			0.48; *p = 0.030*	
Anti-HP0385			0.57			0.41; *p = 0.006*	
Anti-HP0243/NapA			0.63			0.39; *p = 0.016*	
Anti-HP0371/AccB/FabE			0.65			0.37; *p = 0.017*	
Anti-HP0900/HypB			0.50			0.35; *p = 0.048*	
Anti-HP0709			0.61			0.30; *p = 0.003*	
Panel of 11 anti-HP			0.81			NA	
**Early Gastric Cancer (EGC)**							
GKN1 (4.94 ng/mL)	700 (200 H; 140 EGC; 360 advanced GC)	GC vs. H	0.995	91.2	96	NA	Yoon et al., 2019 [50]
		EGC vs. H	1.000	79.3	96		
		advanced GC vs. H	1.000	95.8	96		
CEA + CA724 + IL-6 + IL-8 + TNFa	Discovery: 497 (204 H; 117 AH; 63 EGC; 113 advanced GC)	GC vs. H	0.95	NA	NA	NA	Li et al., 2018 [51]
		EGC vs. H	0.95				
		advanced GC vs. H	0.95				
CA724 + IL-6 + IL-8 + TNFα		GC vs. AH	0.97	NA	NA	NA	
		EGC vs. AH	0.98				
		advanced GC vs. AH	0.96				
CEA + CA724 + IL-6 + IL-8 + TNFα	Validation: 165 (66 H; 41 AH; 19 EGC; 39 advanced GC)	GC vs. H	NA	89.66	92.42	NA	
		EGC vs. H		84.21	90.91		
		advanced GC vs. H		92.31	90.91		
CA724 + IL-6 + IL-8 + TNFα		GC vs. AH	NA	87.93	87.80	NA	
		EGC vs. AH		78.95	85.37		
		advanced GC vs. AH		92.31	90.24		
SPRR2A (80.7 pg/mL)	490 (100 H; 100 CG; 200 GC (I + II *n* = 122, III + IV *n* = 78); 40 RC; 50 CC)	GC (I + II) vs. H	0.78	69.6	68.1	NA	Xu et al., 2020 [52]
ITIH4 (171.2 ng/mL)	400 (178 H; 37 Hpi; 28 LGN; 38 EGC; 70 advanced GC; 49 OST)	EGC vs. H	0.8394	73.08	94.44	NA	Sun et al., 2021 [53]
Signature of 19 proteins: CEACAM5 + CA9 + MSLN + CCL20 + SCF + TGFa + MMP-1 + MMP-10 + IGF-1 + CDCP1 + PPIA + DDAH-1 + HMOX-1 + FLI1 + IL-7 + ZBTB-17 + APBB1IP + KAZALD-1 + ADAMTS-15	150 (50 H; 100 GC (I *n* = 8, II *n* = 20, III *n* = 57, IV *n* = 14, missing *n* = 1))	GC (I + II) vs. H	0.99	89	100	NA	Shen et al., 2019 [54]
TFF3	155 (44 H; 111 GC (I *n* = 42, II *n* = 39, III *n* = 27, IV *n* = 3))	GC (I) vs. H	0.703	83.3	54.5	NA	Choi et al., 2017 [55]
CDH17		GC (II + III) vs. H	0.667	77.3	61.4		
HPT Asn-211: Hex_6_HexNAc_5_Fuc_1_NeuAc_1_ HPT Asn-241: Hex_6_HexNAc_5_Fuc_1_NeuAc_1_Hex_7_HexNAc_6_Fuc_1_	25 (15 H; 10 GC (I *n* = 5, III-IV *n* = 5))	GC (I) vs. H	1	100	100	NA	Lee et al., 2018 [56]
THBS2	120 (41 H; 33 BGT; 46 EGC)	EGC vs. H	0.816	NA	NA	NA	Li et al., 2021 [57]
		EGC vs. BGT	0.840				
CA19-9		EGC vs. H	0.901				
		EGC vs. BGT	0.847				
THBS2/CA19-9		EGC vs. H	0.951				
		EGC vs. BGT	0.928				
THBS1	89 (29 H; 31 EGC; 29 advanced GC)	EGC vs. H	0.646	NA	NA	NA	Yoo et al., 2017 [58]
		advanced GC vs. H	0.656				
Clusterin isoform 1		EGC vs. H	0.878				
		advanced GC vs. H	0.937				
Vitronectin		EGC vs. H	0.756				
		advanced GC vs. H	0.833				
Tyrosine protein kinase SRMS		EGC vs. H	0.887				
		advanced GC vs. H	0.856				
AHSG	Validation: 130 (28 H; 42 GC (I *n* = 5, II *n* = 11, III *n* = 21, IV *n* = 5); 30 CRC; 30 HCC)	GC vs. H	0.93	NA	NA	NA	Shi et al., 2018 [59]
		GC (I + II) vs. H	0.82				
FGA		GC vs. H	0.98				
		GC (I + II) vs. H	0.98				
APOA-1		GC vs. H	0.83				
		GC (I + II) vs. H	0.96				
9 TAA-panel: c-Myc + p16 + HSPD1 + PTEN + p53 + NPM1+ ENO1 + p62 + HCC1.4	814 (407 H; 407 GAC (I *n* = 67, II *n* = 87, III *n* = 142, IV *n* = 40, unknown *n* = 71))	GC vs. H	0.857	71.5	71.3	NA	Qin et al., 2019 [60]
		GC (I + II) vs. H	0.737	64.9	70.5		
	Training: 410 (205 H; 205 GAC (I *n* = 38, II *n* = 42, III *n* = 75, IV *n* = 33; NA *n* = 17))	GC (I + II) vs. H					Yang et al., 2020 [61]
TAA-panel I: p53 + COPB1 + GNAS + PBRM1 + ACVR1B							
		Panel I					
		Training	0.885	66.7	94.6	NA	
		Validation	0.821	76.7	83.3		
TAA-panel II: p53 + SMARCB1 + COPB1 + SRSF2 + GNAS	Validation: 252 (126 H; 126 GAC (I *n* = 15, II *n* = 15, III *n* = 27, IV *n* = 7; NA *n* = 62))						
		Panel II					
		Training	0.869	74.7	90.3	NA	
		Validation	0.876	76.7	80.9		
RAE1 aAbs	364 (122 H; 51 PL; 78 EGC; 113 advanced GC)	EGC vs. H	0.745	94.9	47.1	NA	Zhu et al., 2023 [62]
PGK1 aAbs			0.648	41	87.6		
NPM1 aAbs			0.611	20.5	100		
PRDX3 aAbs			0.613	88.5	65.2		
UBE2N aAbs			0.585	24.4	95		
ARF4 aAbs			0.488	11.5	98.3		
ANXA2 aAbs			0.563	19	86.9		
ANOS1	367 (66 H; 301 GC (I *n* = 225, II *n* = 47, III *n* = 26, IV *n* = 3))	GC vs. H	0.7058	36	85	NA	Kanda et al., 2020 [63]
		GC (I) vs. H	0.7131	36	85		
DPYSL3		GC vs. H	0.6188	48	82		
		GC (I) vs. H	0.5948	45	82		
MAGED2		GC vs. H	0.5031	28	92		
		GC (I) vs. H	0.5113	27	92		
TrxR activity	Before clinical intervention: 261 (130 H; 131 GC (I *n* = 25, II *n* = 39, III *n* = 46, IV *n* = 21)	GC (I) vs. H	0.948	80.00	97.69	NA	Peng et al., 2019 [64]
		GC (II) vs. H	0.955	84.62	97.69		
		GC (III) vs. H	0.971	89.13	96.15		
		GC (IV) vs. H	0.974	90.48	97.69		
		GC vs. H	0.963	85.5	97.69		
CEA + CA19-9 + CA72-4		GC vs. H	0.834	78.41	96.92		
CEA + CA19-9 + CA72-4 + TrxR		GC vs. H	0.982	91.60	94.62		

** AGC, AGC of any grade; * AGA, AGA of any grade; advanced GC, advanced gastric cancer; AG, atrophic gastritis; AG2+, moderate to severe AG; AGA, AG in the antrum; AGA2+, moderate to severe AGA; AGAC, atrophic gastritis of the antrum and corpus; AGAC2+, moderate to severe AGAC; AGC, AG in the corpus; AGC2+, moderate to severe AGC; AH, atypic hyperplasia; AUC, area under the curve; BGT, benign gastric tumor; CAG, chronic atrophic gastritis; CC, colon cancer; CG, chronic gastritis; CI, confidence interval; CRC, colorectal cancer; DYS, dysplasia; EGC, early gastric cancer; GA, gastric atrophy; GAC, gastric adenocarcinoma; GAG, gastric atrophy group; GC, gastric cancer; GIM, gastric intestinal metaplasia; H, healthy controls; HCC, hepatocellular cancer; Hpi, chronic superficial gastritis associated with *Helicobacter pylori* infection; IM, intestinal metaplasia; LGD, low-grade dysplasia; LGN, low-grade intraepithelial neoplasia; NA, not available; NAG, non-atrophic gastritis; OR, odds ratio; OST, other system malignant tumors; PL, precancerous lesions; RC, rectal cancer.

#### 3.1.1. Gastric Physiology-Related Biomarkers

AG affects the gastric corpus, causing a reduction in pepsinogen (PG). PGI is found in chief cells (in the oxyntic mucosa), while PGII has a more widespread distribution; a low ratio of PGI/PGII is therefore a marker of oxyntic/corpus atrophy [65,66]. Low PGI serum levels and PGI/PGII ratios can be used to identify patients with advanced-stage AG [67,68] requiring endoscopy, especially if their *H. pylori* serology results are negative [69]. Based on blood sampling, in the 2000s, it was proposed that the combination of the serum levels of PGI, PGII, and Gastrin-17 (G-17) and anti-*H. pylori* IgG antibodies are reliable parameters with which to diagnose AG [68,70,71], as recently reviewed [32]. The measurement of these blood biomarkers estimated the global prevalence of AG to be between 23.9% and 27.0%, while for the same patients histological analysis showed an AG prevalence of 31.6–33.4% [72]. The corresponding test, referred to as GastroPanel^®^ (Biohit, Helsinki, Finland), allows for the evaluation of the capacity of the corpus and antrum to produce acid and gastrin, respectively, in order to estimate the inflammation of the gastric mucosa as well as the severity and location of AG. More recently, the diagnostic performance of GastroPanel^®^ in the detection of AG was evaluated on 519 Finnish patients with dyspepsia, who were consulting regarding a gastroscopy [37]. AG was detected in 53 patients (10.2%), with a concordance between serological data and the biopsy diagnosis of 92.4% (95% CI = 90.0–94.6), and a weighted kappa of 0.86. Receiver operating characteristic (ROC) curve analysis for the prediction of moderate/severe AG in the corpus (AGC2+) gave area under the curve (AUC) values of 0.954 with a sensitivity of 92% and a specificity of 98.8% for PGI (cut-off value of 30 ng/mL) and an AUC of 0.993 with a sensitivity of 100% and a specificity of 98.6% for the PGI/PGII ratio (cut-off value of 3) (Table 1). In a French multicenter study on 344 patients at high risk of GC having a gastroscopy, a 50% rate was detected for AG. The GastroPanel^®^ diagnostic performance showed a sensitivity of only 39.9% (95% CI = 31.9–48.2), with a specificity of 93.4% (95% CI = 88.9–96.4), a positive predictive value (PPV) of 81.9% (95% CI = 71.1–90.0), and a negative predictive value (NPV) of 67.3% (95% CI = 61.4–72.8). Sensitivity was significantly higher, allowing us to detect severe atrophy 60.8% (95% CI = 46.1–74.6) and corpus AG (AGC) 61.0% (95% CI = 49.2–72.0), but was insufficient to detect either antral or corpus mild atrophy [38]. In that case, the diagnostic performance of GastroPanel^®^ was not statistically different from that of PGI alone [38], and lower compared to that seen in the Finnish study [37]. More recently, a French prospective multicenter study, using ChemiLuminescent Enzyme ImmunoAssay (CLEIA), was reported. This was performed on 356 patients undergoing upper endoscopy, including 91 with non-atrophic gastritis (NAG), 152 with AG, and 113 with normal mucosa (H). Sera were tested for PGI and PGII, in parallel with interleukin-6 (IL-6), human epididymal protein 4 (HE-4), adiponectin, ferritin, and Krebs von den Lungen 6 (KL-6). In comparison to controls (H and NAG), for the detection of moderate to severe corpus AG (AGC2+/AGAC2+), the best specificities are observed at cut-off values for both PGI (≤20.2 ng/mL) and PGI/PGII (≤0.96), 95.6% and 98%, respectively (Table 1). Moreover, for the detection of moderate to severe antrum AG (AGA2+), PG alone was not sufficient, with a level not significantly different compared to controls (H and NAG). Interestingly, IL-6 detects AG in the antrum (AGA2+) with a sensitivity of 72.2% (95% CI = 46.5–90.3). While the combination of PGI/PGII ratio with HE-4 showed a better specificity of 96.1% (95% CI = 92.4–98.3), its sensitivity was only 40.7% (95% CI = 27.6–55.0) for the detection of moderate to severe AG at any stomach location (AG2+) [39]. Thus, the diagnostic performance of PG testing is better in terms of detecting severe atrophy and corpus AG, but less efficient for detection of antral and corpus mild atrophy.

The measure of PGI/PGII ratio has already been proposed in the 2000s for the management of patients with low or high grade IM or DYS [73]. A Malaysian study conducted on 72 patients (48 controls without chronic atrophic gastritis (H), 12 patients with chronic atrophic gastritis (CAG), 9 patients with IM, and 3 patients with GC) analyzed the sensitivity and specificity of serum PGI, PGI/PGII ratio, and G-17 to diagnose AG and IM [40]. Patients with corpus CAG or IM had a lower PGI/PGII ratio compared with the control group (7.2 vs. 15.7, *p < 0.001*). The severity of corpus AG and IM was correlated with the PGI/PGII ratio (r = −0.42, *p < 0.001*), with an optimal cut-off of 10.0 (AUC = 0.902; sensitivity: 83.3%; specificity: 77.9%), thus giving better diagnostic performance than the French cohort [39]. These data support the notion that the PGI/PGII ratio is specifically predictive for corpus CAG and IM, correlated with the type and severity of lesions, at least in Asian populations. More recently, a retrospective Chinese study on EGC patients with low- or high-grade gastric intraepithelial neoplasia (GIN) proposed the use of PGI and G-17 to differentiate GIN according to their grade of severity [74].

#### 3.1.2. Immunity and Related Autoantibodies

Parietal cell autoantibodies (PCAs) are primarily directed against the gastric proton pump H^+^/K^+^ ATPase (ATP4A and ATP4B). Anti-ATP4B autoantibodies (aAbs) are produced during autoimmune gastritis (AIG), induced by *H. pylori* infection, through the molecular mimicry mechanism due to cross-reactivity between bacterial peptides and gastric H^+^/K^+^ ATPase [75,76]. The decrease in ATP4B has been associated with the malignant transformation of the gastric mucosa and GC aggressiveness [77]. PCAs are prevalent in AG patients, thus reflecting atrophic damage to the oxyntic mucosa [78,79]. They are diagnostic markers for AIG and Biermer’s disease, which are characterized by the presence of AG [80].

The Luciferase ImmunoPrecipitation System (LIPS) is a technology that enables the detection of specific serum antibodies via the immunoprecipitation of the corresponding luciferase reporter-tagged antigens. A prospective case-finding study on 218 patients who underwent gastric biopsies reported the measurement of ATP4A and ATP4B aAbs by LIPS, concomitantly with the quantification of global PCAs and PGI serum levels by ELISA [41]. Histopathology was used to classify 107 subjects in the group of patients with AG in the corpus (AGC) (autoimmune 81.2%, and multifocal extensive 18.8%) and 111 subjects in the control group (H). In AGC patients, ATP4A, ATP4B, and PCAs titers were at higher levels than in the controls, whereas PGI serum level was reduced (*p < 0.0001*). ROC-AUC analysis for ATP4A, ATP4B, PCAs and PGI, performed to predict the presence of AGC, indicated sensitivities of 75%, 77%, 69%, and 73% with specificities of 88%, 88%, 91%, and 80%, respectively (Table 1). Thus, PCAs showed a higher specificity than PGI, highlighting their good performance in predicting AGC. In combination with PGI, they should improve AG diagnosis, specially in the corpus.

AG precedes the development of IM (Figure 1B), a pivotal stage in the progression to GC. To assess the individual risk of evolution to cancer, it is mainly recommended to consider IM instead of atrophy [16,46], thus highlighting the need to further characterize IM-specific biomarkers.

### 3.2. Intestinal Metaplasia (IM)

Gastric IM induces histological changes in the gastric glands, such as the acquisition of the characteristics of intestinal glands, the development of a brush border, and the presence of goblet and Paneth cells not usually present in the normal gastric mucosa (Figure 1A(c)). Conventional White-Light endoscopy cannot accurately diagnose IM [25]. Virtual staining techniques are more sensitive, but may be limited due to significant inter-observer variability [81]. Through NBI technology, IM are characterized by the presence of particular features, including tubulo-villous and irregular mucosal patterns, light blue crests, variable vascular density, and white opaque fields, as shown in Figure 1A(d) [82]. IM can be distinguished as complete (type I) and incomplete (type II–III). Patients with incomplete IM are associated with a less differentiated phenotype and a higher grade of severity [83]. A large prospective longitudinal study indicated that, while the difference was not statistically significant, OLGIM stage II–IV patients with incomplete IM were at increased risk of EGC compared with complete IM patients (Hazard Ratio (HR) 5.96; 95%  CI = 0.77–46.4; *p = 0.09*) [46]. Moreover, patients with IM (OLGIM III–IV) were at higher risk of GC within two years (adjusted-HR 20.7; 95% CI = 5.04–85.6; *p < 0.01*) compared to patients with OLGIM II, who were also at significant risk of EGC (adjusted-HR 7.34; 95% CI = 1.60–33.7; *p = 0.02*). Importantly, the progression of IM to higher levels of malignancy can take about 6 years, a period during which the diagnosis of IM and periodical surveillance are essential. A risk-stratified approach has been proposed, recommending endoscopic surveillance for high- (OLGIM III–IV) and intermediate-risk (OLGIM II) patients within 2 and 5 years, respectively [46]. In addition to the severity of IM, the location and patient familial history of GC are determinant criteria in endoscopy surveillance. These data further highlight the importance of liquid biopsy-based biomarker discovery to improve the detection/prevention of gastric cancer in these patients. In line with this, several serum protein candidates have emerged, as listed in Table 1, including inflammation-related factors, autoantibodies, and *H. pylori* antibodies.

#### 3.2.1. Inflammation and Immunity-Related Proteins

Circulating serum levels of inflammatory mediators are a characteristic of tissue inflammation associated with GPNLs. They may reflect gastric tissue activity and are directly correlated with systemic regulation. Using Luminex^®^ bead-based assays, the serum level of 28 immune- and inflammation-related markers was evaluated in 174 *H. pylori*-positive individuals, including 99 patients with IM or DYS and 75 controls with NAG [42]. Five biomarker candidates were identified: two C-C motif ligands (CCL) chemokines/macrophage inflammatory protein (MIP) (CCL3/MIP1A; CCL20/MIP3A) and interleukins IL-1ß, IL-4 and IL-5 (Figure 1B). As reported in Table 1, the degree of association of the third tertile category with IM was similar for CCL3/MIP1A, IL-4, and IL-5, and lower for CCL20/MIP3A and IL-1β. The T-helper 2 (Th2) cytokines IL-4 and IL-5 showed a correlation of 0.5, allowing us to distinguish NAG from IM. IL-1ß is one component of the Th17 immune response and also involves IL-17A induction by *H. pylori* infection. Recently, higher serum IL-17 levels have been reported in *H. pylori*-infected patients with IM and DYS (*n* = 45), compared to NAG patients (*n* = 45) and healthy subjects (H) (*n* = 45) [43]. The ROC analysis for IL-17A showed AUC values of 0.62 and 0.67 for IM/DYS vs. NAG and H (Table 1), respectively, leading the authors to propose IL-17A as a potent biomarker for predicting IM and/or DYS and improving the prevention of GC.

C-reactive protein (CRP) plays an important role in inflammation. It is involved in the production of cytokines such as IL-6. The serum CRP level increases with the degree of inflammation in *H. pylori*-infected patients with chronic gastritis [84]. A study on patients with AG (GAG; *n* = 68), using gastric IM (GIM; *n* = 98) and NAG (H; *n* = 70) patients as controls, investigated serum CRP levels [44]. Both GIM and GAG groups showed serum CRP levels that were 2.5-fold higher than those of the controls. ROC analysis indicated an AUC of 0.77, with a sensitivity of 66.3% and specificity of 77.6%, for differentiating patients with IM and/or AG from those with chronic NAG (Table 1), with a CRP cut-off value ≥1.95 mg/L. In the same study, serum neopterin levels were also measured. Neopterin is a signaling pyrazino-pyrimidine compound, defined as a biomarker of cellular immune responses, with pre-diagnostic capacity in the development of CRC [85]. As with CRP, neopterin serum levels are higher among *H. pylori*-positive patients. Pearson’s correlation analysis confirmed that high serum neopterin levels were positively correlated with the presence of AG and IM, with an AUC of 0.876 (sensitivity: 93%; specificity: 76%), corresponding to a neopterin cut-off value ≥ 10.15 nmole/L.

Also playing important roles in response to gastric mucosal injuries and inflammation are the Trefoil Factor Family (TFF) peptides, strongly expressed in mucus-producing cells. TFFs include three members that are involved in the defense and repair of the mucosa and in tumorigenesis [86]. The progressive loss of TFF1 and TFF2, along with the induction of TFF3, has long been thought to be involved in the early stages of gastric carcinogenesis [87]. At the tissue level, TFF3 is normally absent from the pyloric mucosa, except in IM [88]. In a cross-sectional Latin American study on 288 patients, diagnosed via gastric endoscopy (110 GIM; 14 GC), and 164 H controls, slight but significantly higher median serum TFF3 levels were detected by ELISA in the GIM group (13.1 ng/mL) compared with the controls (11.9 ng/mL) *(p =* 0.024) [45]. Interestingly, an increasing number of OLGIM stages is associated with higher levels of serum TFF3 (Rho coefficient = 0.124, *p* = 0.04). The TFF3 diagnosis performance for GIM is characterized by an AUC of only 0.58 (95% CI = 0.51–0.65), with a moderate sensitivity of 55.5% and a specificity of 58.5%. In order to identify serum biomarkers for high-risk OLGIM, a large prospective longitudinal multicenter cohort study of 2980 patients was carried out in Singapore from 2004 to 2010 [46]. GIM was diagnosed in 1321 patients (44.3%) (OLGIM I *n* = 906; OLGIM II-IV *n* = 415). The serum levels of TFF3, macrophage Migration Inhibitory Factor (MIF), pleiotropic immunoregulatory cytokines, and PGI/PGII were measured. While MIF levels and the PGI/PGII ratio decreased with increasing OLGIM, TFF3 increased and allowed the differentiation of OLGIM 0/II from OLGIM III/IV among *H. pylori*-negative patients (AUC = 0.749; 95% CI = 0.628–0.870; *p < 0.01*), proving to be a higher-performing serum biomarker than PG under these conditions. These data were confirmed by testing a Chinese cohort of nearly 4000 patients, as was the association between TFF and the presence of GPNLs [47]. In this study, patients with IM (*n* = 1002) showed the highest concentration of serum TFF3 levels, with an odds ratio (OR) of 1.92 (Table 1). Not only TFF3, but also serum TFF1 and TFF2 levels increased progressively from CAG to IM, low-grade dysplasia (LGD), and GC. Therefore, it seems that TFF3 performs better as a diagnostic marker in Latin American populations compared with Asian populations.

#### 3.2.2. Immunity and Related Autoantibodies

Serum aAbs against tumor-associated antigens (TAAs) have been proposed as sensitive immunodiagnostic biomarkers, especially for the risk stratification of patients with premalignant lesions. A recombinant antigen microarray was developed to analyze the prevalence of aAbs directed against 102 TAAs in large case-control cohorts from two independent Caucasian and Asian populations [89]. Thirteen and eight aAbs signatures were identified in the Caucasian and Asian cohorts, respectively, allowing GC patients to be distinguished from H controls (Caucasian: sensitivity of 24%—specificity of 91%; Asian: sensitivity of 24%—specificity of 93%). aAbs against cancer/testis antigen 2 (CTAG2), DEAD-box helicase 53 (DDX53), and the artificial peptide ID1625 were detected in both cohorts. These aAbs signatures were then investigated in the serum of 100 CAG and 775 IM Caucasian patients, according to the OLGIM score. Seroreactivity tends to increase with the increase in OLGIM score and was significantly higher in advanced/severe IM (OLGIM III/IV). GC-associated seroreactivity was detected in 13% of patients with advanced/severe IM (OLGIM III/IV) and its expression increased in comparison with mild/moderate IM (5.3%) (OLGIM I/II), comparable to what was seen in EGC patients (12%). Autoantibody reactivity against CTAG2 and DDX53 gave a similar response with stage I GC of the intestinal type, indicating that both OLGIM III/IV and EGC patients have comparable aAb responses. Interestingly, these data indicate that the humoral immune response against TAAs is generated during the earliest stages of GPNLs, as a defense mechanism against cancer cell development.

#### 3.2.3. Antimicrobial Defense

*H. pylori* infection is a major event at the origin of the development of IM [12]. Using Luminex^®^ (Thermo Fischer Scientific, Waltham, MA, USA)-based *H. pylori* multiplex serology [90], serum samples from 1402 individuals of a Chinese high-risk GC population, including 412 patients with IM, 145 with DYS, and 333 with undifferentiated DYS, were assayed for antibodies against 13 *H. pylori* recombinantly expressed fusion proteins (UreA, Catalase, GroEL, NapA, CagA, HP0231, VacA, HpaA, Cad, HyuA, Omp, HcpC and HP0305) [48]. Omp and HP0305 were found to be the strongest markers of risk for the presence of GPNLs with an OR of 5.37 (95% CI = 4.20–6.89) and OR, 3.85 (95% CI = 3.04–4.88), respectively. Moreover, a classification model of GPNLs that included age, smoking, and *H. pylori* seropositivity for Omp and HP0305 resulted in an AUC of 0.751 (95% CI = 0.725–0.777) and OR of 7.43; (95% CI = 5.59–9.88) (see Table 1); these were even better than the AUC value of the *H. pylori* oncoprotein CagA: 0.7184 (95% CI = 0.68–0.74).

More recently, using *H. pylori*–nucleic acid programmable protein arrays (NAPPA), the humoral response to 1528 *H. pylori* proteins was investigated by comparing IM and NAG patients in the discovery (*n* = 50/group) and validation (*n* = 100/group) cohorts. Among the 62 IgG and 11 IgA antibodies with more than 10% seropositivity in IM and/or NAG group, 12 IgG and 6 IgA antibodies showed relative higher seroprevalence in IM than in NAG cases, leading to the identification of a signature of 11 IgG as having the best diagnostic performance to distinguish IM from NAG (AUC = 0.81; 95% CI = 0.75–0.87) (see Table 1). Anti-CagA, previously reported to positively correlate with the presence of IM [91], gave an AUC of 0.77 (95% CI = 0.70–0.84), close to that of anti-Omp27 (AUC = 0.73) [49]. As was also previously reported [48], this signature included *H. pylori* antibodies against outer membrane proteins (OMPs) and essential factors for bacterial survival and gastric colonization (Figure 1B).

### 3.3. Dysplasia (DYS)

Gastric DYS are associated with a high risk of synchronous carcinoma in other areas of the stomach [92], a condition found in up to 30% of patients with gastric DYS [93]. The World Health Organization (WHO) defines DYS as the presence of an histologically unequivocal neoplastic epithelium without evidence of tissue invasion [94] and associated with gastric inflammation [95]. Several DYS classification systems—including the Padova, Vienna, and WHO systems—have been developed to standardize the definition of gastric DYS and neoplasia between Western and Japanese pathologists. Two types of DYS have been described as having gastric and intestinal immunophenotypes associated with low (Figure 1A(e,g)) and high grades (Figure 1A(f)) of severity, respectively [96]. The progression incidences from low-grade (LGD) and high-grade DYS (HGD) to carcinoma were reported to be 2.8–11.5% and 10–68.8%, respectively. A nationwide cohort study in the Netherlands, including patients previously detected for GPNLs who were identified in the Dutch nationwide histopathology registry (PALGA), indicated 10-year GC risks in patients with mild to moderate DYS and severe DYS of 4% and 33%, respectively [97]. Due to the higher risk of progression to carcinoma in HGD patients, endoscopic or surgical resection is recommended [98], while for LGD patients, it is only recommended in the case of visible lesions [16,99]. Because of the rapid progression from HGD to gastric neoplasia, the detection of DYS is challenging. The sensitivity of White-Light endoscopy in detecting DYS is 51–74% and this can be further improved to 92% by magnifying endoscopy with NBI [100]. Importantly, even after endoscopic resection (ER), the long-term and periodical surveillance of gastric DYS is strongly recommended, highlighting the usefulness of non-invasive biomarkers in the follow-up of these patients.

Despite the fact that the identification of liquid biopsy-based biomarkers is crucial in order to detect DYS, they have been investigated in very few studies, likely due to their challenging diagnosis. As cited above, the serum level of IL-17A has been reported to be an indicator of the presence of IM and DYS and a potential biomarker with which to predict GC development [43]. IL-17A production is induced by *H. pylori*. For this, the presence of antibodies against its virulence factors, OMP and HP0305, has been proposed as a serological biomarker, not only in relation to the presence of IM but also that of DYS [48]. This short list, as illustrated in Figure 1B, highlights the special efforts required to improve the diagnosis and early detection of DYS, an essential requirement to reduce the global burden of GC.

### 3.4. Early Gastric Cancer (EGC)

EGC is defined as a kind of gastric carcinoma confined to the mucosa and/or submucosa (T1), irrespective of lymph node involvement (Figure 1A(h)) [101,102,103]. When the tumor invades the muscularis propria (T2), it is classified as advanced GC. The vessel plus surface (VS) endoscopic classification system has been described as being able to differentiate between cancerous and non-cancerous lesions using magnifying endoscopy [104]. Its robustness in diagnosing EGC is well validated [105,106]. VS classification evaluates the microvascular (MV) and micro-surface (MS) patterns. The diagnostic criteria are (1) the presence of an irregular MV pattern with a demarcation line and (2) the presence of an irregular MS pattern with a demarcation line. Another criterion to be considered is the absence of glands, with complete architectural loss of the mucosal and vascular pattern predicting neoplastic changes in the gastric mucosa (Figure 1A(i)) [16]. Lesions that meet at least one of these criteria are considered to be cancerous, with a sensitivity of 97% [107]. This Japanese classification is not easily transposable to GC low-incidence countries, where endoscopic screening at the population scale is not pertinent, virtual chromoendoscopy is not systematically performed, and magnification is not always available.

In contrast to DYS, circulating plasma/serum proteins to diagnose EGC have been abundantly investigated, as indicated by the numerous biomarker candidates discussed in the recent literature (Figure 1B). In the present review, only studies that clearly distinguish EGC from advanced GC and that define the diagnostic performance of the identified proteins have been considered.

#### 3.4.1. Gastric Physiology-Related Proteins

Due to its role in gastric mucosal defense through the regulation of the NFkB signaling pathway and cytokine expression, Gastrokin 1 (GKN1) is a significant player in the GC process [108]. GKN1 is a stomach-specific protein that is highly expressed in normal gastric mucosa. It is located in the superficial/foveolar gastric epithelium in the antrum, body, and cardia. GKN1 expression is decreased in gastric tumor tissues and derived cell lines [109]. Serum GKN1 level was measured by ELISA in 500 GC patients from South Korea, among which 140 had EGC and 360 had advanced GC. It was significantly lower compared to the value in H controls (*n* = 200). ROC analysis showed that a GKN1 cut-off value of 4.94 ng/mL clearly discriminated GC patients from controls with a high AUC of 0.995 (95% CI = 0.9919–0.9988), a sensitivity of 91.2%, and a specificity of 96%. Moreover, the serum GKN1 level was lower in advanced GC cases than in EGC patients, allowing these patients to be distinguished from controls, with the best AUC value of 1.0. The sensitivity was 79.3% and specificity of 96% for EGC and 95.8% and 96% for advanced GC patients, respectively (Table 1) [50]. Importantly, serum GKN1 concentrations measured in seven other types of cancer, including hepatocellular carcinoma (HCC), colorectal (CRC), breast (BRC), and ovarian (OVC) and prostate (PRC) cancers, did not show significant differences compared with controls, supporting GKN1 as a potent, specific diagnostic biomarker for EGC and advanced GC.

#### 3.4.2. Inflammation and Immunity-Related Biomarkers

Interleukin-26 (IL-26), belonging to the IL-10 cytokine family, is produced by Th17 cells. IL-26 regulates chronic inflammation processes and autoimmune disease [110]. The serum concentration of IL-26, CEA, CA19-9, CA125, CA72-4, and ferritin was measured by ELISA in 100 patients with benign gastric diseases and 302 GC patients, including stages I (*n* = 75), II (*n* = 73), III (*n* = 125) and IV (*n* = 29) [111]. Serum IL-26, CEA, CA19-9, CA125, and CA72-4 levels were positively correlated with the severity of gastric lesions and were differentially significant among the 5 groups of patients (r = 0.528, *p < 0.001*; r = 0.314, *p < 0.001*; r = 0.236, *p = 0.017;* r = 0.197, *p = 0.032*; r = 0.285, *p < 0.001*, respectively). In contrast, ferritin is negatively correlated with the severity of GC lesions (r = −0.329; *p = 0.015*).

Luminex bead-based assays were developed and used on a discovery cohort of 497 individuals (63 EGC, 113 advanced GC, 117 atypical hyperplasia (AH), and 204 H controls) to measure serum CEA and CA72-4 levels in combination with serum IL-6, IL-8, and TNFα levels, leading to the proposal of a diagnostic model [51]. ROC analysis established an AUC of 0.95 (95% CI = 0.93–0.97) to discriminate between H and GC patients and 0.95 (95% CI = 0.92–0.98) to discriminate between H and EGC or advanced GC patients. Interestingly, the combination CA72-4, IL-6, IL-8, and TNFα gave better AUC values of 0.97 (95% CI = 0.95–0.99), 0.98 (95% CI = 0.96–0.99) and 0.96 (95% CI = 0.94–0.98) to discriminate between AH and GC, and EGC and advanced GC, respectively [51]. A joint analysis performed on a validation cohort of 165 individuals (66 H, 41 AH, 19 EGC, and 39 advanced GC patients) confirmed that the proposed models discriminate EGC patients from H subjects using the combination of CEA + CA72-4 + IL-6 + IL-8 + TNFα, with a sensitivity of 84.21% and specificity of 90.91%, whilst the combination CA72-4 + IL-6 + IL-8 + TNFα, with a sensitivity of 78.95% and specificity of 85.37%, discriminated EGC patients from patients with AH. Thus, the panel of these inflammatory mediators may provide a potent screening tool with which to detect EGC lesions (Figure 1B).

The Small Proline-Rich Protein 2A (SPRR2A) has been recently identified as a novel target for p73; it is a member of the p53 tumor suppressor family and may contribute to inflammation [112]. The diagnostic performance of SPRR2A was investigated by ELISA in serum samples from 100 controls (H), 100 patients with chronic gastritis (CG; 48 chronic superficial gastritis and 52 CAG), 200 with GC (I + II *n* = 122; III + IV *n* = 78), 40 with rectal cancer (RC), and 50 with colon cancer (CC) [52]. The correlation between serum SPRR2A levels, GC clinical pathological parameters, and ROC analysis was considered. The median serum SPRR2A concentration in GC patients was significantly higher than in controls and gastritis or CC patients (*p < 0.001*). A cut-off value of 80.7 pg/mL yielded an AUC of 0.851 (95% CI = 0.785–0.916; sensitivity: 75.7%; specificity: 74.5%) and 0.820 (95% CI = 0.742–0.899; sensitivity: 90.5%; specificity: 61.7%), discriminating GC patients from controls and from gastritis patients, respectively. However, when distinguishing GC patients at stage I and II from controls, the AUC for the serum SPRR2A was a little bit lower, 0.78 (95% CI = 0.669–0.891; sensitivity: 69.6%; specificity: 68.1%), indicating that SPRR2A is not among the best EGC biomarkers.

Most of the best-known serological cancer biomarkers are glycoproteins, such as CA19-9, CEA, CA15-3, and CA79-9, which are also related to inflammation. Despite their current use in clinical oncology, their predictivity of EGC and/or GC lesions is low. Recent studies have further identified glycoprotein candidates for the prediction of EGC, among which the glycoprotein inter-alpha-trypsin heavy chain 4 (ITIH4), belonging to the inter-alpha-trypsin inhibitor (ITI) family, shows a good performance in cancer prediction [113]. ITIH4 is a type II acute phase protein involved in inflammatory host responses to trauma, closely related to tumorigenesis and metastasis. Combining several methods (mass spectrometry, ELISA, Western blot (WB), and immunohistochemical staining), the serum ITIH4 level was evaluated in a Chinese population cohort of 400 individuals. Patients presented lesions of chronic superficial gastritis (CSG), associated with *H. pylori* infection (Hpi; *n* = 37); low-grade intra-epithelial neoplasia (LGN), corresponding to the precancerous group (*n* = 28), EGC (*n* = 38); advanced GC (*n* = 70); and other system malignant tumors (OSTs) (*n* = 49). H individuals (*n* = 178) were also included as controls [53]. For all cases, the diagnosis was confirmed via a combination of upper gastrointestinal endoscopy, magnifying endoscopy narrow-band imaging (ME-NBI), endoscopic ultrasonography, and histopathology. Using mass spectrometry analysis, significantly higher levels of ITIH4 were observed in serum samples from EGC patients compared to advanced GC and H, with a high diagnostic performance corresponding to an AUC of 0.839 (95% CI = 0.7393–0.9396) at a cut-off level of 171.2 ng/mL, with a sensitivity of 73.08% and specificity of 94.44% used to discriminate EGC cases from controls.

Protein combinations allow for a higher diagnostic performance than single biomarkers. In a retrospective study based on the recruitment of 100 GC patients, including 28 with EGC (TNM I-II stage), and 50 H individuals [54], high-throughput protein detection technology, using multiplex proximity extension assays (PEAs), identified over 300 proteins, and a signature of 19 serum proteins that together distinguished GC cases from controls. They included carcinoembryonic antigen-related cell adhesion molecule 5 (CEACAM5 or CEA), carbonic anhydrase 9 (CA9), mesothelin (MSLN), C-C motif chemokine 20 (CCL20), stem cell factor/KIT ligand (SCF), transforming growth factor alpha (TGF-a), matrix metalloproteinase-1 (MMP-1), matrix metalloproteinase-10 (MMP-10), insulin-like growth factor I (IGF-1), CUB domain-containing protein 1 (CDCP1), peptidyl-prolyl cis-trans isomerase A (PPIA), dimethylarginine dimethylaminohydrolase 1 (DDAH-1), heme oxygenase 1 (HMOX-1), friend leukemia integration 1 transcription factor (FLI1), IL-7, zinc finger and BTB domain-containing protein 17 (ZBTB-17), amyloid beta A4 precursor protein-binding family B member 1-interacting protein (APBB1IP), kazal-type serine protease inhibitor domain-containing protein (KAZALD-1), and a disintegrin and metalloproteinase with thrombospondin motifs 15 (ADAMTS-15). They are related to inflammation and/or the immune response (IL-7, PPIA, HMOX-1, ZBTB-17, APBB11P, CCL20), metabolism and cellular physiology (CA9, IGF-1, DDAH-1, FLI1), cell cycle regulation (TGFa), cell adhesion (CEACAM5, MSLN, CDCP1), cell differentiation (SCF), and the extracellular matrix (MMP-1, MMP-10, KAZALD, ADAMTS-15). The variation of each protein was analyzed by univariate analysis. Elastic-net logistic regression was performed to select serum proteins for the diagnostic model. Together, these proteins provided an increased diagnostic capacity to discriminate EGC patients at TNM stages I-II (AUC = 0.99; sensitivity: 89%; specificity: 100%) from H controls, compared to the consideration of each protein separately. The best diagnostic performance for a single protein of this panel was achieved for MMP-1, with an AUC of 0.75, a sensitivity of 68%, and a specificity of 78% [54].

As mentioned above, TFFs are also related to the inflammatory process, and their use was previously proposed to improve GC screening [114]. Using ELISA, Choi et al. measured the levels of TFF3 and cadherin 17 (CDH17) related to cell adhesion. As with TFF3, CDH17 is recognized as a tissue marker for IM [115]. The analysis was carried out on plasma samples from 111 GC patients and 44 H individuals. The GC group includes 42, 39, 27, and 3 cases related to TNM stages I, II, III, and IV, respectively [55]. Both plasma CDH17 and TFF3 levels were increased in GC patients compared to controls. TFF3 levels were significantly different between GC stage I (9.913 ± 0.841 ng/mL) and H (6.195 ± 0.702 ng/mL) (*p = 0.001*) and CDH17 levels between GC stages II (0.578 ± 0.091 ng/mL) and III (0.549 ± 0.088 ng/mL) and H samples (0.329 ± 0.060 ng/mL) (*p = 0.023* and *0.037*, respectively). As reported in Table 1, ROC analysis for CDH17 (GC stages II-III), performed to differentiate between GC stages and controls, gave an AUC of 0.667 (*p = 0.003*), with a sensitivity of 77.3% and specificity of 61.4%, and ROC analysis for TFF3 (GC stage I) gave a higher AUC of 0.703 (*p = 0.001*), with a sensitivity of 83.3% and specificity of 54.5% [55].

Haptoglobin (HPT) is one of the major acute phase glycoproteins, accounting for 0.4% to 2.6% of blood proteins. The aberrant glycosylation of HPT has been associated with chronic inflammation and cancer [116]. A targeted glycoproteomic platform using nanoliquid chromatography (LC)/quadrupole time-of-flight (Q-TOF) mass spectrometry (MS) and MS/MS, combined with antibody-assisted purification, was set up to investigate specific glycan structures and the involvement of HPT glycosylation in GC [56,117]. Sera from 15 H controls and 10 GC patients, subdivided into two groups based on TNM classifications (stage I *n* = 5 and stage III-IV *n* = 5), were tested. After HPT pronase digestion, fingerprint glycopeptides (glycan moiety + small peptide tag) representing each glycosite were quantitatively monitored for the efficient tracking of site-specific glycoform changes in HPT: HN dipeptide, NHSE tetrapeptide, NAT, and HPN tripeptides that were selected as peptide tags for glycosites Asn-184, Asn-207, Asn-211, and Asn-241, respectively. The greatest magnitude of difference was observed at Asn-241, and the most significant difference was seen at Asn-211 where fucosylated complex-type glycans were found to be 9.6-fold and 4.2-fold more abundant in GC than in H patients (*p = 6.06 × 10^−5^* and *p = 2.2 × 10^−7^*, respectively). Finally, based on ROC analyses (AUC = 1; sensitivity: 100%; specificity: 100%), three fucosylated and/or sialylated complex-type glycans were identified as potential biomarkers: we tested Hex_6_HexNAc_5_Fuc_1_NeuAc_1_ at Asn-211, Hex_6_HexNAc_5_Fuc_1_NeuAc_1_ at Asn-241, and Hex_7_HexNAc_6_Fuc_1_ at Asn-241. When testing only EGC, these three complex-type glycans still corresponded to AUC = 1 [56]. Although further investigations are required for these data to be confirmed as valid, for example, using larger cohorts and considering patients with GPNLs, protein glycosylations constitute promising biomarkers with which to detect EGC.

Another example is thrombospondins (THBSs), belonging to Ca^2+^-binding glycoproteins, secreted from immune and mesenchymal cells, and endotheliocytes. Through interactions with a large range of proteins, THBSs are implicated in various biological procedures, including cell-to-cell and cell-to-matrix interactions, cell migration, blood vessels production, apoptosis, and cytoskeletal regulation. Serum THBS2 and CA19-9 levels were measured by ELISA in blood samples from 41 H individuals, 33 benign gastric tumor (BGT) patients, and 46 EGC patients. The benign or EGC stages were confirmed according to the American Joint Committee on Cancer (AJCC) TNM (tumor–node–metastasis) classification [57]. The serum THBS2 level in EGC and BGT patients was upregulated dramatically compared to H individuals (*p < 0.05*), as was the level of CA19-9 (*p < 0.05*). A significant correlation between THBS2 and CA19-9 serum levels was observed only in EGC patients (*p = 0.04*), and the methods showed a good capacity to distinguish EGC from H with AUC values of 0.816 (95% CI = 0.722–0.911) and 0.901 (95% CI = 0.833–0.968), respectively. Furthermore, the combination of both enhanced their predictivity, with an individual index of AUC = 0.951 (95% CI = 0.912–0.989). Thus, THBS2 or CA19-9 were able to predict EGC as single biomarkers and their combination improved their diagnostic performance (Figure 1B).

In a Korean study including 60 GC patients (31 EGC and 29 advanced GC) and 29 H controls, THBS1 with clusterin isoform 1, vitronectin, and tyrosine-protein kinase SRMS were also identified as potent GC biomarkers by quantitative mass spectrometry (MS/MS) [58]. ROC analysis indicated AUCs of 0.646, 0.878, 0.756, and 0.887 for THBS1, clusterin isoform 1, vitronectin, and tyrosine protein kinase SRMS, respectively, for the discrimination of EGC from controls. In the case of advanced GC, the diagnostic accuracy is better for clusterin isoform 1, with an AUC of 0.937 while AUC values were 0.833, 0.856, and 0.656 for vitronectin, Tyrosine protein kinase SRMS, and THBS1, respectively.

High-throughput proteomic technologies, such as magnetic-bead-based purification and matrix-assisted laser desorption/ionization time-of-flight mass spectrometry, were applied to serum samples from 32 GC patients (both pre- and post-operatively) and 30 H volunteers, leading to the identification of 12 peptide candidates. Overall, 10 of the peptides corresponded to 6 proteins: isoform I of fibrinogen alpha chain precursor (FGA), alpha-2-HS-glycoprotein precursor (AHSG), apolipoprotein A-I precursor (APOA-I), hemoglobin subunit beta (HBB), cytoskeleton-associated protein 5 (CKAP5), and eukaryotic peptide chain release factor GTP-binding subunit ERF3B (GSPT2) [59]. Based on these data, a validation cohort including 42 paired GC patients—(pre- and post-operative samples) among which 16 and 26 were at stages I/II and III/IV, respectively, alongside 30 CRC and 30 HCC patients and 28 H volunteers—was used to evaluate the serum level of these candidates by ELISA. This study further confirmed the diagnostic accuracy of FGA, AHSG, and APOA-1, with significantly higher amounts detected, specifically in GC patients versus H controls, with AUCs of 0.98 (95% CI = 0.95–1.00), 0.93 (95% CI = 0.87–0.99) and 0.83 (95% CI = 0.73–0.93), respectively. Importantly, EGC and AGC stages could be distinguished, with significantly higher levels of FGA, AHSG and APOA-1 in GC stage I/II compared with H controls (Figure 1B), with AUCs of 0.98 (95% CI = 0.96–1.01), 0.82 (95% CI = 0.69–0.95), and 0.96 (95% CI = 0.91–1.01), respectively.

#### 3.4.3. Immunity and Related Autoantibodies

Serum aAbs against tumor-associated antigens (TAAs), reported above as IM biomarkers, are also able to distinguish EGC patients [89]. In a case-control study on 407 GC patients in the gastric adenocarcinoma group (GAC) (I *n* = 67; II *n* = 87; III *n* = 142; IV *n* = 40; unknown *n* = 71) and 407 H controls, aAbs against 14 TAAs were measured by ELISA [60]. A panel of 9 aAbs against TAAs, including c-Myc, p16, HSPD1 (Heat Shock Protein Family D (Hsp60) Member 1), PTEN (Phosphatase and tensin homolog), p53, NPM1 (Nucleophosmin 1), ENO1 (Enolase 1), p62, and HCC1.4, was identified, and it could distinguish GC cases from H controls with an AUC of 0.857 (sensitivity: 71.5%; specificity: 71.3%). Interestingly, this panel also identified EGC cases (stages I/II) from H with an AUC of 0.737 (sensitivity: 64.9%; specificity: 70.5%). The production of these aAbs could promote the risk of GC and GC aggressiveness, as their presence is associated with a worse prognosis.

Anti-p53 has also been reported as a potent EGC biomarker, with 4 other aAbs against TAAs (Panel I: anti-COPB1, anti-GNAS, anti PBRM1, anti-ACVR1B or Panel II: anti-SMARCB1, anti-COPB1, anti-SRSF2, anti-GNAS), in a study on independent training (205 GAC and 205 H) and validation (126 GAC and 126 H) cohorts, according to an immunodiagnostic prediction model using logistic regression (LR) and Fisher linear discriminant analysis (LDA), respectively [61]. For the training cohort, the diagnostic accuracy of these panels when distinguishing EGC (stages I + II) patients led to AUCs of 0.885 (sensitivity: 66.7%; specificity: 94.6%) and 0.869 (sensitivity: 74.7%; specificity: 90.3%) for panels I and II, respectively. The analysis of the validation cohort showed higher sensitivity of 76.7% but lower specificities of 83.3 to 80.9% for panels I and II, respectively.

Using serological proteome analysis (SERPA) associated with nanoliter-liquid chromatography combined with quadrupole time-of-flight tandem mass spectrometry (Nano-LC-Q-TOF-MS/MS), 7 aAbs corresponding to RAE1 (mRNA export factor 1), PGK1 (phosphoglycerate kinase 1), NPM1 (nucleophosmin 1), PRDX3 (thioredoxin-dependent peroxide reductase), UBE2N (ubiquitin-conjugating enzyme E2), ARF4 (ADP-ribosylation factor 4) and ANXA2 (annexin A2) have also been reported as being able to identify patients with precancerous lesions (PL) and EGC [62]. The aAbs were tested on 364 serum samples from 242 patients (51 PL, 78 EGC, 113 advanced GC) and 122 controls (H) for their ability to detect precancerous lesions and GC by ELISA. All of the aAbs were present at higher levels in patients with PL, EGC, and AGC than in H. Anti-RAE1 best discriminated GC patients at different stages, with AUCs of 0.710 (95% CI = 0.628–0.793), 0.745 (95% CI = 0.678–0.811), and 0.804 (95% CI = 0.750–0.858) for PL, EGC, and advanced GC, respectively. The AUC was also calculated for panels incorporating multiple aAbs, for PL, EGC, and advanced GC, showing that a combination of 3 aAbs (RAE1, NPM1, and PGK1; Model 1) has a slightly increased AUC compared to RAE1 aAb alone. The use of two predictive models—considering gender, RAE1, PGK1, NPM1, and ARF4 aAbs (Model 2 for PL) and age, gender, RAE1, PGK1, and NPM1 aAbs (Model 3 for EGC)—improved diagnostic efficiency, with AUCs of 0.803 (95% CI = 0.736–0.860) and 0.857 (95% CI = 0.800–0.902), sensitivities of 66.7% and 75.6%, and specificities of 78.7% and 87.7%, respectively. These values are higher than the diagnostic accuracy obtained using a single index.

#### 3.4.4. Cellular Physiology and Metabolism Related Proteins

In a prospective multicenter study, the diagnostic performances of the secreted glycoprotein anosmin 1 (ANOS1), a component of the extracellular matrix; of the dihydropyrimidinase-like 3 (DPYSL3), a cell adhesion molecule involved in metastasis; and of MAGED2, related to the melanoma-associated antigen (MAGE) family involved in cancer development, were evaluated. Sera were collected from 66 H volunteers and 301 GC patients, who were classified into four groups according to the criteria of the 7th edition of the Union for International Cancer Control (UICC): I *n* = 225 (74%); II *n* = 47 (16%); III *n* = 26 (9%); and IV *n* = 3 (1%), [63]. The serum levels of ANOS1, DPYSL3 and MAGED2 were quantified by ELISA. ANOS1 showed the highest AUC value (0.7058) for the discrimination of patients with GC from H. However, sensitivity and specificity for ANOS1 were 36% and 85%, respectively, compared to 48% and 82% for DPYSL3, and to 28% and 92% for MAGED2. Among the 301 GC patients, the correlation coefficients of serum levels for ANOS1/DPYSL3, DPYSL3/MAGED2, and MAGED2/ANOS1 were 0.4698, 0.2318, and 0.5095 (*p < 0.0001*), respectively, indicating modest correlation between each pair. When evaluating their capability to discriminate patients with GC stage I (*n* = 225) from H, the AUC values for ANOS1, DPYSL3, and MAGED2 were 0.7131, 0.5948, and 0.5113, respectively. The levels of ANOS1 were significantly elevated in patients with stage I GC compared with H controls (median 1179 ng/mL and 461 ng/mL, respectively, *p < 0.0001*), whereas they were equivalent in patients with GC stages I and II–IV.

Mammalian thioredoxin reductase (TrxR) is a selenium-containing oxidoreductase that catalyzes the NADPH-dependent reduction of thioredoxin (Trx) disulfide and participates in several redox-sensitive signaling cascades that mediate numerous physiological processes. Trx was highly expressed in various malignancies and cancers. In a Chinese study, the diagnostic efficacy of TrxR activity, measured in vitro the by 5, 5′-dithiobis (2-nitrobenzoic) acid (DTNB) reduction assay, was compared with the concentrations of well-known GC biomarkers, analyzed by Electrochemiluminescence Immunoassay (ECLIA) [64]. A total of 923 patients, including 131 with GC before clinical intervention (I *n* = 25; II *n* = 39; III *n* = 46; IV *n* = 21), 662 with GC after chemical drug treatment (I *n* = 40; II *n* = 148; III *n* = 179; IV *n* = 295) (staged according to the 8th IASLC/AJCC staging system), and 130 H controls, were enrolled. The plasma TrxR activity [median (IQR)] in GC patients before clinical interventions [9.09 (7.96, 10.45) U/mL] was significantly higher (*p < 0.0001*, Mann–Whitney U test) than in H controls [3.69 (2.38, 5.32) U/mL]. The critical value of TrxR activity for GC diagnosis was set at 7.34 U/mL with an AUC of 0.963 (95% CI = 0.943–0.983; sensitivity of 85.50%; specificity of 97.69%). The combination of CEA, CA19-9 and CA72-4 exhibited an improved diagnostic efficacy for GC cases (AUC 0.834; 95% CI = 0.778–0.891; sensitivity: 78.41%; specificity: 96.92%) relative to any individual biomarker (*p < 0.05*). Notably, when adding TrxR activity to this panel, diagnostic performance for GC was further improved with an AUC of 0.982 (95% CI = 0.970–0.993), sensitivity of 91.6%, and specificity of 94.62%. Consistent with previous studies, serum CEA, CA72-4, and CA19-9 levels remained slightly altered in phase I/II GC patients compared with H controls. Importantly, plasma TrxR activity in phase I/II GC patients was significantly higher than in H controls (*p < 0.0001*), highlighting its high sensitivity and strong diagnostic performance (AUC > 0.900) in EGC diagnosis.

Thus, a wide panel of biomarkers has been identified to distinguish EGC from H controls and from AGC (Figure 1B). Importantly, a panel of biomarkers is associated with a better diagnostic accuracy than any single biomarker, as supported by various studies [54,61], reflecting the complexity of the mechanisms associated with the early steps of gastric carcinogenesis.

## 4. Discussion and Future Perspectives

GC is mostly associated with bad prognosis due to its asymptomatic phenotype until it reaches an advanced stage, thus highlighting the need for its primary prevention and early detection. The identification of blood biomarkers appears to be the most suited strategy to improving the screening of patients at risk of GC. This is crucial from the perspective of large-scale preventive strategies. Due to their accessibility, stability, and easy quantification, blood proteins are a valuable source of biomarkers for the diagnosis, prognosis, and monitoring of cancer. More than 100 United States Food and Drug Administration-cleared or -approved biomarkers are plasma proteins [118]. Recent technical advances in protein analysis and omics technologies have been adapted, combining immunodetection-based assays such as ELISA together with Luminex-based multiplex assays and antibody arrays coupled to mass spectrometry-based high-throughput methods, and have largely improved protein biomarker discovery. Importantly, most studies show that protein biomarker panels are more representative of the cellular physiology status and associated with a better diagnostic accuracy than a single candidate. The current challenge is to identify the most robust and promising combination that will be easily translated to clinical use [119]. This requires the standardization of the preanalytical handling of blood-based samples [120] and statistical methods.

An important highlight of the present review is the lack of blood protein biomarkers to predict the presence of DYS. Apart from a few inflammation-related proteins [42,43] and anti-*H. pylori* factors [48], no specific candidates for low- or high-grade DYS have been reported. Due to the high GC risk associated with DYS, further studies and clinical trials should be developed to overcome this drastic shortcoming. As described previously, the fleeting nature of DYS makes the endoscopic detection and identification of related biomarkers challenging. The future improvement of endoscopy technologies and omics global approaches coupled to multiplex technologies should overcome this gap.

Autoimmune gastritis has been proposed as a preneoplastic step associated with an annual incidence of GC of 0.3% [121]. The presence of PCAs in 20 to 30% of *H. pylori*-positive patients and of anti-*H. pylori* antibodies in AIG patients further supports the link between *H. pylori* infection and AIG. An important highlight of this review is the identification of aAbs as promising biomarker candidates for AG, IM, and EGC detection. Compared to the use of PGI for the detection of AG, aAbs showed better diagnostic performance [41]. Cancer cell-induced immunological responses can result in the production of aAbs against TAAs and are emerging candidates to predict EGC lesions, for example, as panels including aAbs against tumor suppressors such as p53 and p16 or the oncogene cMyc [60]. Moreover, aAbs have been reported to distinguish late-stage IM in both Caucasian and Asian populations, for example, in the use of aAbs anti-CTAG2 and anti-DDX3 in GC stage I patients [89]. These variations in serum aAbs support the increasing evidence that immune dysregulation and autoimmunity impact the development of gastric neoplasia, as does *H. pylori*-induced autoimmune gastritis [122]. Importantly, some recent epidemiological data suggest that there is an increasing incidence in some young patient groups, possibly due to autoimmunity; if this tendency is confirmed, it may change the epidemiology of GC in the future. In these cases, the main event inducing the gastric carcinogenesis process would not be *H. pylori*-induced chronic inflammation [123]. In this hypothesis, aAbs could be more robust biomarkers that are less impacted by the physiopathological primum movens.

Our review further confirms the robustness of long-known GPNL biomarkers. The loss of gastric glands that characterize AG is associated with variation in the serum PGI level and PGI/PGII ratio, which also correlates with the severity of AG and IM [37,39]. Also related to inflammation, the measurement of serum TFF3 in combination with PG has been proposed as an alternative to endoscopy for the early detection of gastric lesions. Data from studies in Singaporean [46], Chinese [47], and Chilean [45] populations confirmed that TFF3 is a potent serum biomarker that can identify patients with gastric IM and distinguish low (OLGIM 0-II)- from high-grade (OLGIM III–IV) IM [46]. However, the best diagnostic performance in detecting IM is related to antimicrobial defense, as observed with a panel of 11 antibodies directed against *H. pylori* bacterial factors that discriminate IM vs. NAG patients [49]. Although this panel is promising, it cannot be generalized to all populations as it does not take into account the difference in the immunogenic profile among races/ethnicities, and it also excludes patients previously eradicated for *H. pylori* or those that never had the infection.

EGC biomarkers are the most investigated. In addition to serum aAbs, inflammation and immunity-related proteins are potent biomarkers with which to predict EGC lesions. The use of high-throughput technologies led to the identification of promising panels with better diagnostic accuracy than single proteins, as illustrated by a signature of 19 proteins showing 100% specificity in terms of distinguishing GC (stages I and II) from H subjects, despite its characterization using a small size cohort [54]. Related to inflammation, the combination of IL-6, IL-8, TNFα, and CA72-4 led to the discrimination of EGC, with one of the best diagnostic performances [51], as also observed using THBS2 combined with CA19-9 [57]. CA19-9, CEA, and CA72-4 are all serum tumor markers for the diagnosis of GC [124]. Their combination with the measurement of Trx activity increases the diagnosis accuracy of each component of this panel in predicting GC. It should be noted that the measurement of Trx activity alone is associated with a specificity of 97% for the discrimination of EGC stage I and II lesions from H subjects [64]. Further validation on a larger scale and on multicentric cohorts taken from various populations would be worthwhile to further validate this signature. Post-translational modifications (PTMs), such as protein glycosylation, are also promising biomarkers [56], with the identification of three HPT fucosylated and/or sialylated complex-type glycans associated with 100% sensitivity and specificity in terms of predicting EGC lesions. However, as for other studies, these PTMs were identified in a small cohort and need further validation.

Together, these results show that subtle alterations in the gastric mucosa result in sufficient changes to be detected in blood. This can be explained by (a) the sensitivity of detection techniques (b) the specificity of the normal gastric mucosa cell types and derived compounds, since their modifications are easily detectable. An important parameter to take into account in future investigations is related to the distinct epidemiological, clinical, and pathological characteristics between Asian and non-Asian populations. As the results of data obtained in Asian populations are not systematically transposable to non-Asian populations, these data must be interpreted with caution. Further analysis and validation studies with a special focus on patients from restricted geographical areas, such as our recent PREGASIGN prospective and multicentric study that aims to identify specific plasma protein biomarkers of GPNLs and EGC in the French population (ClinicalTrials.gov Identifier: NCT05854368), are required, together with assessments of cohorts from populations of various geographic origins, before translation for clinical use. These studies could pave the way for the development of reliable non-invasive diagnostic tests that could be useful in the context of large-scale population screening.

## 5. Conclusions

In conclusion, liquid biopsy-based biomarkers constitute a crucial tool that can reduce the global burden of GC of both intestinal and diffuse subtypes. Their discovery offers a new prospect for the screening, early detection, and monitoring of individuals at risk of developing GC. An important gap is the lack of studies aiming to specifically identify biomarkers for IM and more importantly DYS. Further studies and clinical trials should be developed to achieve breakthroughs in the field.

Most of the studies focused on Asian populations, potentially due to the high GC prevalence in these countries; however, the diagnostic potency of identified biomarkers can differ according to the considered population due to the multifactorial origin of this cancer.

Future research should consider the standardization of the preanalytical handling of blood-based samples [120] and statistical methods, together with adequate reporting, the design of sufficiently large multicenter studies, and independent study populations, to facilitate the comparability of results and the demonstration of both the clinical validity and clinical utility, the first step in the clinical adoption of a liquid biopsy test.

## Figures and Tables

**Figure 1 cancers-16-03019-f001:**
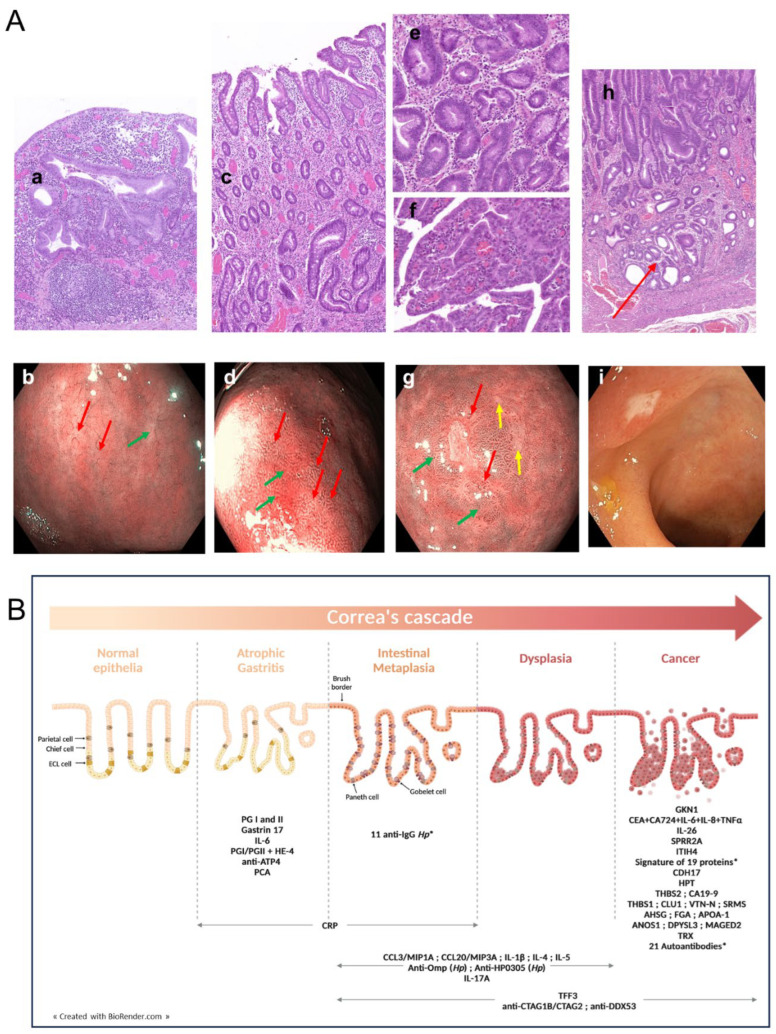
(**A**) Histologic and endoscopic characteristics of gastric lesions during gastric carcinogenesis. Gastric histologic lesions obtained from gastric biopsies, H&E staining (a,c,e,f,h) and examination in NBI (b,d,g) and White Light. (i) Olympus endoscopy without optical zoom. Lesions of gastric atrophy (a,b). a (17.4×): chronic follicular atrophic non-metaplastic gastritis (complete atrophy). b: glandular atrophy shows pale appearance of gastric mucosa (green arrow), increased visibility of vasculature due to thinning of the gastric mucosa (red arrows), and loss of gastric folds. Lesions of intestinal metaplasia (c,d). c (9.4×): chronic atrophic and metaplastic gastritis (severe atrophy, severe intestinal metaplasia). d: intestinal metaplasia (red arrows) in gastric corpus. Rounded to branched pit pattern, with mucosa slightly darker than adjacent mucosa. Green arrows show normal gastric corpus mucosa, with round pit pattern. Lesions of dysplasia (e,f,g). e (31.5×): low-grade dysplasia. f (31.3×): high-grade dysplasia. g: endoscopically visible low-grade dysplasia with central ulcer. Vessel disorganization (red arrows) and glandular disorganization (green arrows) are observed. Glandular atrophy (yellow arrows) surrounds the lesion. Lesions of early gastric cancer (h,i). h (3.8×): early gastric cancer (intramucosal, red arrow). i: early gastric cancer centered on an ulcer. Absent glands with complete architectural loss of the mucosal and vascular pattern. (**B**) Plasma biomarkers identified at each step of the gastric cancer cascade. * detailed in Table 1.

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
