# Peer review of "Protein Biomarkers of Gastric Preneoplasia and Cancer Lesions in Blood: A Comprehensive Review"

_cancers, 2024, doi:10.3390/cancers16173019_

Round 1

Reviewer 1 Report

Comments and Suggestions for Authors

The authors have reviewed the status of protein biomarkers of gastric preneoplasia and cancer in blood.

The manuscript is comprehensive and ambitious and the topic is of high clinical and scientific interest. In particular, the Introduction is clear and well written. 

Table 1 seems to belong close to section 3 rather than being followed by section 2, which describes endoscopic diagnosis, before section 3 again returns to blood-based markers. 

In Table 1, the contents are comprehensive and abbreviations are numerous. Although formally correct, the table is indeed very hard to read. AGC and AGC+ are also two different abbreviations, can this be avoided? Can some abbreviations be omitted? GIMG? Both PGL and PL are used, but are there also precancerous lesions not being gastric? I would also advice against using Se and Sp as unconventional abbreviations in the table and in Section 3.1.1 in the manuscript as well.  In the first paper in Table  1 (Koivurova) they also examined the Gastropanel, but these data are not presented in the table?

It should be mentioned that the relative proportion of Lauren intestinal type cancers has declined considerably paralleling the H pylori prevalence and that new biomarkers in low-endemic countries (the West) must include markers of diffuse type cancers as well. Perhaps most markers in the Asian studies referred to in Table 1 are markers related to intestinal metaplasia and atrophy. Please reflect and comment in the manuscript, is this a knowledge gap? Have any specific markers of diffuse type cancer been identified in the mentioned publications? 

Figure 1 b,d,g and i have poor resolution. Do the authors have more illustrative images with zoom, NBI, BLI or what it takes to convey the message?

Lines 196-8. Please clarify that PGI is found in chief cells (in the oxyntic mucosa) while PGII has a more widespread distribution and a low ratio is therefore a marker of oxyntic / corpus atrophy. Refer e.g. to the original publications by Samloff 1971 and 1973. PMID: 4935210   PMID: 4124404

In the section 3.2. please note that intestinal metaplasia (IM) in the antrum is a weak risk factor for cancer compared to IM in the corpus / fundus to the extent that surveillance in not recommended (PMID: 30841008, with literature review).

In section 3.4.1 the Gastrokin1 is mentioned. Please provide some information about the localization, cell type, in tumour? In precancerous mucosa? 

The important distinction in the beginning of the Introduction between non-cardia and cardia subtypes and histological type (Laurens intestinal type being H pylori related) is not maintained throughout the manuscript, perhaps for a good reason. Do some or none of the publications try to differ between these distinct entities (histologically and etiologically) when results are presented? Combining different entities is tempting to increase n, but in biomarker studies the sensitivity, performance and AUC may thereby be reduced. 

Author Response

The manuscript is comprehensive and ambitious and the topic is of high clinical and scientific interest. In particular, the Introduction is clear and well written. 

Table 1 seems to belong close to section 3 rather than being followed by section 2, which describes endoscopic diagnosis, before section 3 again returns to blood-based markers. 

We thank the reviewer for this comment. The table 1 has been shifted at a more appropriate place, after the section 2. 

In Table 1, the contents are comprehensive and abbreviations are numerous. Although formally correct, the table is indeed very hard to read. AGC and AGC+ are also two different abbreviations, can this be avoided? Can some abbreviations be omitted? GIMG? Both PGL and PL are used, but are there also precancerous lesions not being gastric? I would also advice against using Se and Sp as unconventional abbreviations in the table and in Section 3.1.1 in the manuscript as well. 

We thank the reviewer for this comment. Accordingly, for a better understanding AGC has been replaced by « advanced GC » both in the Table 1 and the text. GIMG was used to reference to the original publication as PGL and PL. Both in the Table 1 and the text, GIMG that referred to gastric intestinal metaplasia group has been replaced by GIM and PGL used for precancerous gastric lesions has been replaced by PL for precancerous lesions.  Also abbreviations Se and Sp has been replaced by sensitivity and specificity, respectively.

In the first paper in Table  1 (Koivurova) they also examined the Gastropanel, but these data are not presented in the table?

Data presented in the paper from Koivurova et al reported separately the diagnostic accuracy of the different biomarkers as gastrin-17 (G-17), PGI and PGI/PGII to distinguish the gastric lesions, as documented in the table 1. In this study, no diagnostic values are reported concerning their combination referred as the Gastropanel assay.  

It should be mentioned that the relative proportion of Lauren intestinal type cancers has declined considerably paralleling the H pylori prevalence and that new biomarkers in low-endemic countries (the West) must include markers of diffuse type cancers as well. Perhaps most markers in the Asian studies referred to in Table 1 are markers related to intestinal metaplasia and atrophy. Please reflect and comment in the manuscript, is this a knowledge gap? Have any specific markers of diffuse type cancer been identified in the mentioned publications? 

According to the reviewer suggestion, the following sentence has been included lines 60-61 in the introduction: “It concerns intestinal type GC, in parallel to the decrease of H. pylori infection prevalence during this period.”. We fully agree with the comment concerning new biomarkers that must include markers of diffuse type cancer as well. To the best of our knowledge, there are no blood markers associated with diffuse type GC, whose carcinogenesis process appears to be faster than that of the intestinal type and largely unexplored. Moreover, the involvement of pre-cancerous lesions is only mentioned in few studies and not confirmed. Related to the studied populations, there are more Asian studies than Western ones. This is probably because the incidence of GC is higher in this part of the world. It could be a cause of knowledge gap. Most of the Asian studies reported in the Table 1 are related to EGC biomarkers. Only 3 studies among 15 mention the GC histological subtype (Xu et al, 2020; Choi et al, 2017 and Yoo et al, 2017), without a comparison of the efficacy of biomarkers to distinguish between intestinal and diffuse GC type. To the best of our knowledge no specific diffuse GC protein blood biomarkers have been reported in the mentioned publications. This gap has been highlighted in this revised version through the sentence: GC biomarkers discovery should be further investigated in the case of diffuse GC subtype, which remains underexplored as compared to the intestinal subtype (lines 102-103).  

Figure 1 b,d,g and i have poor resolution. Do the authors have more illustrative images with zoom, NBI, BLI or what it takes to convey the message?

We thank the reviewer for this comment. We drastically increased the resolution of the whole Figure 1.

Lines 196-8. Please clarify that PGI is found in chief cells (in the oxyntic mucosa) while PGII has a more widespread distribution and a low ratio is therefore a marker of oxyntic / corpus atrophy. Refer e.g. to the original publications by Samloff 1971 and 1973. PMID: 4935210   PMID: 4124404

We thank the reviewer for this comment. The sentence “PGI is found in chief cells (in the oxyntic mucosa) while PGII has a more widespread distribution and a low ratio is therefore a marker of oxyntic / corpus atrophy” has been included (lines 191-193) and the related references cited.

In the section 3.2. please note that intestinal metaplasia (IM) in the antrum is a weak risk factor for cancer compared to IM in the corpus / fundus to the extent that surveillance in not recommended (PMID: 30841008, with literature review).

Targeted endoscopic surveillance for IM is based on OLGIM stages. As agree with the reviewer comment, focal IM in the antrum is actually a lesion with less potential for degeneration than IM in the fundus. However, it should be emphasised that severe and extensive IM of the antrum is a risk factor for cancer associated with OLGIM grade IV. So on, more than the location of the metaplasia, its severity is a negative prognostic factor. Moreover, according to European recommendations reported in Pimentel-Nunes et al, 2019, “In patients with IM at a single location but with a family history of gastric cancer, or with incomplete IM, endoscopic surveillance with endoscopy in 3 years’ time may be considered”.

The following sentence has been included lines 298-299-300: “In addition to the severity of IM, location and patient familial history of GC are determinant criteria for their endoscopy surveillance”

In section 3.4.1 the Gastrokin1 is mentioned. Please provide some information about the localization, cell type, in tumour? In precancerous mucosa? 

As recommended by the reviewer, information about Gastrokin1 function, localization and expression have been included in section 3.4.1 (lines 471-474)

The important distinction in the beginning of the Introduction between non-cardia and cardia subtypes and histological type (Laurens intestinal type being H pylori related) is not maintained throughout the manuscript, perhaps for a good reason. Do some or none of the publications try to differ between these distinct entities (histologically and etiologically) when results are presented? Combining different entities is tempting to increase n, but in biomarker studies the sensitivity, performance and AUC may thereby be reduced. 

The distinction between non-cardia and cardia subtypes was not maintained throughout the manuscript as it is not considered in the blood marker studies of which we are aware. This may be explained by the difficulty to visualize endoscopically early cardia cancer and, even more so, precancerous lesions in this region. Only few studies reported in our review precise the GC subtypes. As example in the case of EGC detection, among the 15 studies listed in the table 1 only 3 (Xu et al, 2020; Choi et al, 2017 and Yoo et al, 2017) precise the GC histological subtype, but none compared the efficacy of biomarkers to distinguish between intestinal and diffuse GC type.

Reviewer 2 Report

Comments and Suggestions for Authors

The review "Protein biomarkers of gastric preneoplasia and cancer lesions in blood: a comprehensive review" by Thomas Bazin, Karine Nozeret, Catherine Julié, Dominique Lamarque and Eliette Touati is a comprehensive scientific review devoted to the consideration of protein markers for the diagnosis and prognosis of pathological conditions of the stomach from atrophic gastritis to early gastric cancer. The review describes the possibilities of modern instrumental methods for the diagnosis of atrophic gastritis, intestinal metaplasia, dysplasia and gastric cancer. A detailed description allows defining the limitations of diagnostic clearly and confirms the need for non-invasive markers to solve the problems of timely and accurate diagnosis of gastric pathology and assessment of the risk of malignant lesions development.

The diagnostic capabilities were visualized by Figure 1 (page 117). However, Figure 1B is of poor quality and practically unreadable, although it is necessary for a better understanding of the conditions for protein markers application. The review considers various types of markers in chapters devoted to various gastric lesions (Atrophic gastritis, Intestinal Metaplasia, Dysplasia, Early gastric cancer). All the described markers are presented in a table, which, despite its size and a large amount of information, greatly facilitates understanding and systematizes the described markers by groups and states. The review contains a detailed chapter Discussion and future perspectives, in which the authors summarize and outline further areas of the research. The review is well written and contains the newest information on the problem of protein markers for gastric cancer diagnosis. Therefore, I recommend the review for publication in the Cancers journal after minor revision.

Author Response

Comments and Suggestions for Authors

The review "Protein biomarkers of gastric preneoplasia and cancer lesions in blood: a comprehensive review" by Thomas Bazin, Karine Nozeret, Catherine Julié, Dominique Lamarque and Eliette Touati is a comprehensive scientific review devoted to the consideration of protein markers for the diagnosis and prognosis of pathological conditions of the stomach from atrophic gastritis to early gastric cancer. The review describes the possibilities of modern instrumental methods for the diagnosis of atrophic gastritis, intestinal metaplasia, dysplasia and gastric cancer. A detailed description allows defining the limitations of diagnostic clearly and confirms the need for non-invasive markers to solve the problems of timely and accurate diagnosis of gastric pathology and assessment of the risk of malignant lesions development.

The diagnostic capabilities were visualized by Figure 1 (page 117). However, Figure 1B is of poor quality and practically unreadable, although it is necessary for a better understanding of the conditions for protein markers application.

The review considers various types of markers in chapters devoted to various gastric lesions (Atrophic gastritis, Intestinal Metaplasia, Dysplasia, Early gastric cancer). All the described markers are presented in a table, which, despite its size and a large amount of information, greatly facilitates understanding and systematizes the described markers by groups and states. The review contains a detailed chapter Discussion and future perspectives, in which the authors summarize and outline further areas of the research. The review is well written and contains the newest information on the problem of protein markers for gastric cancer diagnosis. Therefore, I recommend the review for publication in the Cancers journal after minor revision.

We thank the reviewer for her/his positive and constructive comments on our review. As she/he suggested, the resolution of Figure 1B has been increased for a better readability.